# Glycosphingolipids Recognized by *Acinetobacter baumannii*

**DOI:** 10.3390/microorganisms8040612

**Published:** 2020-04-23

**Authors:** Miralda Madar Johansson, Mehjar Azzouz, Beatrice Häggendal, Karin Säljö, Henri Malmi, Anton Zaviolov, Susann Teneberg

**Affiliations:** 1Institute of Biomedicine, Department of Medical Biochemistry and Cell Biology, Sahlgrenska Academy, University of Gothenburg, SE40530 Göteborg, Sweden; miralda.madar@gmail.com (M.M.J.); mehjar.azzouz@gu.se (M.A.); beahaggendal@hotmail.com (B.H.); 2Institute of Clinical Sciences, Department of Plastic Surgery, Sahlgrenska Academy, University of Gothenburg, SE41345 Göteborg, Sweden; karin.saljo@vgregion.se; 3Joint Biotechnology Laboratory, MediCity, Faculty of Medicine, University of Turku, FI25020 Turku, Finland; hsmmal@utu.fi (H.M.); anton.zavialov@utu.fi (A.Z.)

**Keywords:** microbial adhesion, *Acinetobacter baumannii*, glycosphingolipid structure, mass spectrometry, human skin glycosphingolipids

## Abstract

*Acinetobacter baumannii* is an opportunistic bacterial pathogen associated with hospital-acquired infections, including pneumonia, meningitis, bacteremia, urinary tract infection, and wound infections. Recognition of host cell surface carbohydrates plays a crucial role in adhesion and enables microbes to colonize different host niches. Here the potential glycosphingolipid receptors of *A. baumannii* were examined by binding of ^35^S-labeled bacteria to glycosphingolipids on thin-layer chromatograms. Thereby a selective interaction with two non-acid glycosphingolipids of human and rabbit small intestine was found. The binding-active glycosphingolipids were isolated and, on the basis of mass spectrometry, identified as neolactotetraosylceramide (Galβ4GlcNAcβ3Galβ4Glcβ1Cer) and lactotetraosylceramide (Galβ3GlcNAcβ3Galβ4Glcβ1Cer). Further binding assays using reference glycosphingolipids showed that *A. baumannii* also bound to lactotriaosylceramide (GlcNAcβ3Galβ4Glcβ1Cer) demonstrating that GlcNAc was the basic element recognized. In addition, the bacteria occasionally bound to galactosylceramide, lactosylceramide with phytosphingosine and/or hydroxy fatty acids, isoglobotriaosylceramide, gangliotriaosylceramide, and gangliotetraosylceramide, in analogy with binding patterns that previously have been described for other bacteria classified as “lactosylceramide-binding”. Finally, by isolation and characterization of glycosphingolipids from human skin, the presence of neolactotetraosylceramide was demonstrated in this *A. baumannii* target tissue.

## 1. Introduction

*Acinetobacter baumannii* is emerging as a worldwide problem as a nosocomial pathogen in hospitalized patients. These bacteria primarily cause pneumonia, but they are also frequent causes of wound and burn infections, bacteremia, meningitis, urinary tract infections, and skin and soft tissue infections. The mortality associated with these infections is high. Isolates resistant to almost all available antimicrobials have been found, thus limiting treatment options. In fact, *A. baumannii* has been classified as highest priority on the recently published WHO list of pathogens needing research and development of new antibiotics [1].

The ability of *A. baumannii* to survive for an extended period of time on artificial surfaces allows it to persist in the hospital environment. This is due to its ability to form biofilms [2]. Biofilm formation in *A. baumannii* is phenotypically associated with exopolysaccharide production and pilus formation [3]. The Csu pili are required for biofilm formation in *A. baumannii* but do not play a role in adherence to human epithelial cells [4]. Recently, the X-ray structure of the CsuC–CsuE chaperone–adhesin complex demonstrated that the tip protein CsuE has three hydrophobic finger-like loops [5]. This unique structural feature mediates bacterial adhesion to abiotic substrates.

Several different factors have been shown to be involved in the adherence of *A. baumannii* to human epithelial cells, as, e.g., the outer membrane protein A (OmpA), the biofilm-associated protein (BAP), the BAP-like proteins 1 and 2 (BLP-1 and BLP-2), the predicted pili subunit encoded by the *LH92_11085* gene, and *Acinetobacter* trimeric autotransporter adhesin (Ata) [6,7,8,9,10]. Several of these factors (OmpA, BAP, BLP-1, BLP-2, and the LH92_11085 gene product) are also involved in biofilm formation, in line with the association found between biofilm production and human epithelial cell adherence [11,12].

Thus, a substantial amount of information about the *A. baumannii* factors involved in adherence to abiotic surfaces and epithelial cells is available. However, much less is known regarding the factors of epithelial cells that the bacteria bind to. Binding to specific receptors on the target cells allows microorganisms to colonize and cause infection and leads to an efficient delivery of virulence factors. The majority of microbial attachment sites on host cells and tissues identified are glycoconjugates [13,14]. In the present study, the potential carbohydrate recognition by *A. baumannii* bacterial cells was investigated by binding of *A. baumannii* bacteria to glycosphingolipids from various sources on thin-layer chromatograms.

## 2. Materials and Methods

### 2.1. A. baumannii Strains, Culture Conditions, and Labeling

*A. baumannii* strains CCUG 890, CCUG 60611, CCUG 68164, and CCUG 19096 were obtained from Culture Collection University of Gothenburg (CCUG). The bacteria were cultured aerobically on blood agar plates and were radiolabeled by the addition of 50 μCi ^35^S-methionine (PerkinElmer; NEG77207MC) diluted in 0.5 mL phosphate-buffered saline (PBS), pH 7.3, to the culture plates. After incubation for 12 h at 37 °C under aerophilic conditions, the bacteria were harvested, centrifuged three times with PBS, and thereafter suspended in PBS containing 2% (*w*/*v*) bovine serum albumin, 0.1% (*w*/*v*) NaN_3_, 0.1% (*w*/*v*) Tween 20, and 1% (*w*/*v*) mannose (BSA/PBS/TWEEN/MANNOSE) to a bacterial density of 1 × 10^8^ CFU/mL. The specific activity of the suspensions was approximately 1 cpm per 100 bacteria.

### 2.2. Reference Glycosphingolipids

Total acid and non-acid glycosphingolipid fractions were isolated as described [15]. Pure reference glycosphingolipids were isolated by repeated chromatography on silicic acid columns and by HPLC and identified by mass spectrometry [16,17] and ^1^H-NMR spectroscopy [18].

### 2.3. Chromatogram Binding Assays

Thin-layer chromatography was done on aluminum-backed silica gel 60 high-performance thin-layer chromatography plates (Merck; 105641/105547). Glycosphingolipid mixtures (20–40 μg), or pure glycosphingolipids (1–4 μg), were applied to the plates, and eluted with chloroform/methanol/water (60:35:8, by volume). Chemical detection was done with anisaldehyde [19].

Binding of radiolabeled bacteria to glycosphingolipids on thin-layer chromatograms was done as described [20]. Dried chromatograms were dipped in diethylether/*n*-hexane (1:5 *v*/*v*) containing 0.5% (*w*/*v*) polyisobutylmethacrylate for 1 min. The chromatograms were blocked with BSA/PBS/TWEEN for 2 h at room temperature. Then the plates were incubated for 2 h at room temperature with ^35^S-labeled bacteria (1–5 × 10^6^ cpm/mL) diluted in BSA/PBS/TWEEN/MANNOSE. After washing six times with PBS and drying, the plates were autoradiographed for 12–36 h using XAR-5 X-ray films (Carestream; 8941114).

Chromatogram binding assays with ^125^I-labeled *Erythrina crista-galli* lectin and *A. baumannii* CsuC–CsuE protein [5] were done as described [21].

### 2.4. Isolation of the A. baumannii Binding Tetraglycosylceramide from Human Small Intestine

A non-acid glycosphingolipid fraction (20 mg) from a human small intestine from our glycosphingolipid collection was first separated by chromatography on a 4 g Iatrobeads (6RS-8060, Iatron Laboratories Inc., Tokyo, Japan;) column eluted with chloroform/methanol/water 60:35:8 (by volume), 27 × 1 mL. The fractions obtained were analyzed by thin-layer chromatography and anisaldehyde staining, and the *A. baumannii* binding activity was assessed using the chromatogram binding assay. The fractions were pooled according to the mobility on thin-layer chromatograms and their *A. baumannii* binding activity. This resulted in an *A. baumannii* binding fraction (8.4 mg) containing mono- to tetraglycosylceramides, which was further separated on a 2 g Iatrobeads column eluted with chloroform/methanol/water 65:25:4 (by volume), 18 × 1 mL. Pooling of the *A. baumannii* binding subfractions gave a fraction (1.7 mg), which was separated on a 1 g Iatrobeads column eluted with chloroform/methanol/water 60:35:8 (by volume), 10 × 0.5 mL. This gave a fraction containing the *A. baumannii* binding tetraglycosylceramide (1.4 mg). This fraction was designated fraction HI-4.

### 2.5. Isolation of the A. baumannii Binding Tetraglycosylceramide from Rabbit Intestine

The non-acid glycosphingolipid fraction (34 mg) from rabbit intestine from our glycosphingolipid collection was separated in a similar manner. The first separation was done on a 2 g Iatrobeads column eluted with chloroform/methanol/water 60:35:8 (by volume), 24 × 1 mL. *A. baumannii* binding fractions were pooled (giving 6.3 mg) and then further separated on a 2 g Iatrobeads column eluted with chloroform/methanol/water 65:25:4 (by volume), 17 × 1 mL. Thereby an *A. baumannii* binding tetraglycosylceramide fraction (1.1 mg) was obtained (designated fraction RI-4).

### 2.6. Isolation of Glycosphingolipids from Human Skin

Full-thickness skin grafts were collected with ethical approval (Dnr-624-16; decision 2016-11-20) after informed consent, at the Department of Plastic Surgery, Sahlgrenska University Hospital. The material was lyophilized and acid and non-acid glycosphingolipids were thereafter isolated as described [15]. Briefly, the material (24.2 g dry weight) was extracted in two steps in a Soxhlet apparatus with chloroform and methanol (2:1 and 1:9, by volume, respectively). The material thereby obtained was subjected to mild alkaline hydrolysis and dialysis, followed by separation on a silicic acid column. Acid and non-acid glycosphingolipid fractions were obtained by chromatography on a DEAE-cellulose column. In order to separate the non-acid glycosphingolipids from alkali-stable phospholipids, the non-acid fraction was acetylated and separated on a second silicic acid column, followed by deacetylation and dialysis. Final purifications were done by chromatographies on DEAE-cellulose and silicic acid columns. Thereby, 28 mg of total non-acid glycosphingolipids was obtained. The total non-acid glycosphingolipid fraction was separated into subfractions by chromatography on a 1 g Iatrobeads column eluted with increasing amounts of methanol in chloroform (10 mL of 10% of methanol in chloroform (by volume), 10 mL of 25% of methanol in chloroform, 10 mL of 33% methanol in chloroform, 10 mL of 75% methanol in chloroform, and 10 mL of 100% methanol). Thereby five glycosphingolipid containing fractions were obtained which were denoted fractions S-1–S-5.

### 2.7. Endoglycoceramidase Digestion and Liquid Chromatography/Electrospray Ionization Mass Spectrometry

Endoglycoceramidase II from *Rhodococcus* spp. (Takara Bio Europe S.A., Gennevilliers, France) was used for hydrolysis of glycosphingolipids, and the oligosaccharides obtained were analyzed by liquid chromatography–electrospray ionization–mass spectrometry (LC-ESI/MS) [17]. In brief, the oligosaccharides were separated on a column (200 × 0.180 mm) packed in-house with 5 μm porous graphite particles (Hypercarb, Thermo Fischer Scientific, Waltham, MA, USA) and eluted with an acetonitrile gradient (A: 8 mM ammonium bicarbonate; B: 100% acetonitrile). The saccharides were analyzed in the negative ion mode on an LTQ linear quadrupole ion trap mass spectrometer (Thermo Electron, San José, CA, USA). The IonMax standard ESI source on the LTQ mass spectrometer was equipped with a stainless-steel needle kept at −3.5 kV. Compressed air was used as a nebulizer gas. The heated capillary was kept at 270 °C, and the capillary voltage was −50 kV. Full-scan (*m/z* 380–2000, 2 microscans, maximum 100 ms, target value of 30,000) was performed, followed by data dependent MS^2^ scans of the three most abundant ions in each scan (2 microscans, maximum 100 ms, target value of 10,000). The threshold for MS^2^ was set to 500 counts. Normalized collision energy was 35%, and an isolation window of 3 u, an activation q = 0.25, and an activation time of 30 ms, was used. The conditions for MS^3^ and MS^4^ were the same, except that the thresholds were set to 300 and 100 counts, respectively.

### 2.8. Inhibition Experiments

*A. baumannii* 890 cells were preincubated with PBS buffer or 1:100 dilutions of antibodies raised against the N-terminal domain of CsuE (αEN) [5] for 2 h, and then assayed for binding to pure reference glycosphingolipids on thin-layer chromatograms, as described above.

## 3. Results

### 3.1. Screening for A. baumannii Carbohydrate Recognition

In the initial screening for carbohydrate recognition by *A. baumannii*, mixtures of glycosphingolipids from various sources were used in order to expose the bacteria to a large number of variant carbohydrate structures. Thus, the binding of the *A. baumannii* strains to acid and non-acid glycosphingolipid mixtures isolated from the small intestine of different species (human, rat, cat, rabbit, dog, monkey, and pig), erythrocytes of different species (human, cat, rabbit, dog, horse, chicken, and sheep), human cancers (lung cancer, kidney cancer, colon cancer, liver cancer, and gastric cancer), and rabbit thymus was tested. There was no binding of *A. baumannii* to any acid glycosphingolipids. However, in most non-acid glycosphingolipid fractions a binding of the bacteria to compounds migrating in the mono- and diglycosylceramide regions was observed (exemplified in Figure 1, lanes 1–4). In addition, a number of more slow-migrating glycosphingolipids in the non-acid fractions of rabbit small intestine, rabbit thymus, rat intestine, and human small intestine were distinctly recognized by the bacteria (Figure 1, lanes 1–4).

During these initial experiments, we also examined the potential carbohydrate binding of *A. baumannii* CsuC–CsuE protein [5], using ^125^I-labeled protein in chromatogram binding assays. However, no glycosphingolipid binding was observed.

### 3.2. Isolation of A. baumannii Binding Tetraglycosylceramides from Human and Rabbit Small Intestine

The *A. baumannii* binding compounds migrating in the tetraglycosylceramide region from rabbit and human small intestine (Figure 1, lanes 1 and 4) were both isolated by chromatography on Iatrobeads columns. The fractions obtained were examined for *A. baumannii* binding on thin-layer chromatograms and pooled according to the mobility on thin-layer chromatograms and their *A. baumannii* binding activity. After several separation steps, 1.1 mg of the *A. baumannii* binding tetraglycosylceramide fraction from rabbit intestine (designated fraction RI-4) (Figure 2A,B, lane 6), and 1.4 mg of the *A. baumannii* binding tetraglycosylceramide fraction from human intestine (designated fraction HI-4) (Figure 2C,D, lane 4), were obtained.

Aliquots of fractions HI-4 and RI-4 were digested with endoglycoceramidase II from *Rhodococcus* spp., and the oligosaccharides obtained were analyzed by LC-ESI/MS, as described below.

### 3.3. LC-ESI/MS of the Human Small Intestinal Glycosphingolipid Recognized by A. baumannii

LC-ESI/MS of oligosaccharides using porous graphitized carbon (PGC) columns gives resolution of isomeric oligosaccharides, and the carbohydrate sequence can be deduced from series of C-type fragment ions obtained by MS/MS (MS^2^) [17,22,23]. Furthermore, MS^2^ spectra of oligosaccharides with a Hex or HexNAc substituted at C-4 have diagnostic cross-ring ^0,2^A-type fragment ions (^0,2^A and ^0.2^A-H_2_O), which allow differentiation of linkage positions. Thus, such ^0,2^A-type fragment ions are found in MS^2^ spectra of oligosaccharides with globo (Galα4Gal) or neolacto/type 2 (Galβ4GlcNAc) core structures but are absent in the spectra obtained from oligosaccharides with isoglobo (Galα3Gal) or lacto/type 1 (Galβ3GlcNAc) core chains.

The base chromatogram from LC-ESI/MS of fraction HI-4A had molecular ion at *m/z* 706, demonstrating an oligosaccharide with one HexNAc and three Hex. MS^2^ of this ion (Figure 3A) gave a C-type fragment ion series (C_2_ at *m/z* 382 and C_3_ at *m/z* 544) identifying an oligosaccharide with Hex-HexNAc-Hex-Hex sequence. The spectrum had a distinct ^0,2^A_2_-H_2_O fragment ion at *m/z* 263 and a ^0,2^A_3_ fragment ion at *m/z* 281, demonstrating a terminal Hex-HexNAc sequence with a 4-substitution of the HexNAc, i.e., a type 2 chain [17,22,23]. The ^0,2^A_4_ ion at *m/z* 646 and the ^0,2^A_3_-H_2_O ion at *m/z* 628 were most likely derived from cross-ring cleavages of the 4-substituted Glc of the internal lactose (Galβ4Glc) part. Taken together, a neolacto tetrasaccharide (Galβ4GlcNAcβ3Galβ4Glc) was thus tentatively identified.

### 3.4. LC-ESI/MS of the Rabbit Small Intestinal Glycosphingolipid Recognized by A. baumannii

The base chromatogram from LC-ESI/MS of fraction RI-4 also had a molecular ion at *m/z* 706 demonstrating an oligosaccharide with one HexNAc and three Hex. MS^2^ of this ion (Figure 3B) also gave a C-type fragment ion series (C_2_ at *m/z* 382 and C_3_ at *m/z* 544) identifying a Hex-HexNAc-Hex-Hex sequence. However, this MS^2^ spectrum had a prominent D_1–2_ ion at *m/z* 202, which is obtained by a C_2_-Z_2_ double cleavage, and diagnostic for a 3-substituted HexNAc [22]. No ^0,2^A_2_-H_2_O fragment ion at *m/z* 263, or ^0,2^A_3_ fragment ion at *m/z* 281, indicating 4-substitution of the HexNAc were present. The ^0,2^A_4_-H_2_O fragment ion at *m/z* 628 and the ^0,2^A_4_ fragment ion at *m/z* 646 were derived from cross-ring cleavages of the 4-substituted Glc of the lactose at the reducing end. No ^0,2^A_3_ fragment ion at *m/z* 484 was present, demonstrating that the Gal of the lactose unit was 3-substituted. Taken together, these spectral features identified a lacto tetrasaccharide (Galβ3GlcNAcβ3Galβ4Glc).

Thus, the human small intestinal glycosphingolipid recognized by *A. baumannii* was tentatively identified as neolactotetraosylceramide (Galβ4GlcNAcβ3Galβ4Glcβ1Cer), while the *A. baumannii* binding glycosphingolipid of rabbit small intestine was tentatively identified as lactotetraosylceramide (Galβ3GlcNAcβ3Galβ4Glcβ1Cer).

### 3.5. Binding to Reference Glycosphingolipids

Thereafter the binding of *A. baumannii* to defined amounts of a number of reference glycosphingolipids structurally related to neolactotetraosylceramide and lactotetraosylceramide was evaluated. The results are exemplified in Figure 4 and summarized in Table 1. Thereby we found that *A. baumannii* bound to lactotriaosylceramide (Figure 4E,F, lane 2), in addition to neolactotetraosylceramide and lactotetraosylceramide. The binding of the bacteria was abolished by substitution of neolactotetraosylceramide or lactotetraosylceramide with an αFuc in the 2-position of the terminal Gal (creating the H type 2 (Figure 4A,B, lane 1, and Figure 4E,F, lane 4) or H type 1 (Figure 4A,B, lane 5) pentaosylceramides). The same effect was obtained by an αFuc in the 3- or 4-position of the internal GlcNAc position (creating the Le^a^ pentaosylceramide (Figure 4A,B, lane 6) or the Le^x^ pentaosylceramide (no. 12 in Table 1). Substitution in the 3-position of the terminal Gal of neolactotetraosylceramide with an αGal (Figure 2A,C, lane 4), or a Neu5Ac or Neu5Gc (Figure 4C,D, lanes 1–3, 5) also abolished the binding. Thus, the minimal requirement for *A. baumannii* glycosphingolipid binding is a GlcNAc, which may be substituted with Galβ3 (lactotetraosylceramide) or Galβ4 (neolactotetraosylceramide). However, further substitutions of the Galβ3GlcNAc or Galβ4GlcNAc sequences abrogates the binding of the bacteria.

Further binding assays using reference glycosphingolipids showed that *A. baumannii* also bound to galactosylceramide (Figure 5A–E, lane 1), lactosylceramide with phytosphingosine and/or hydroxy fatty acids (lane 2), isoglobotriaosylceramide (no. 28 in Table 1), gangliotriaosylceramide (lane 3), and gangliotetraosylceramide (no. 7 in Table 1). While the binding of *A. baumannii* to lactotriaosylceramide, neolactotetraosylceramide, and lactotetraosylceramide was highly reproducible, binding to these compounds was only occasionally obtained. This is exemplified in Figure 2B where galactosylceramide (lane 2) and gangliotetraosylceramide (lane 7) were non-binding, and the bacteria only bound to gangliotriaosylceramide (lane 3), isoglobotriaosylceramide (lane 5), and lactotetraosylceramide from rabbit intestine (lane 6). Binding to lactosylceramide, isoglobotriaosylceramide, gangliotriaosylceramide, and gangliotetraosylceramide has previously been described for other bacteria classified as “lactosylceramide-binding” [14]. The binding of *A. baumannii* to fast-migrating compounds in glycosphingolipid mixtures (exemplified in Figure 1B,C) is thus most likely due to recognition of galactosylceramide and lactosylceramide.

When the relative affinity of *A. baumannii* for the binding-active glycosphingolipids was estimated by binding to serial dilutions of glycosphingolipids on thin-layer chromatograms the bacteria bound to gangliotriaosylceramide, neolactotetraosylceramide, and lactotetraosylceramide with a detection limit of 1–2 μg, while for galactosylceramide 4 μg was required for binding to occur. Thus, there was no clear preferential binding to any of these glycosphingolipids.

It should be noted that the binding patterns obtained with the four *A. baumannii* strains differed to some extent (exemplified in Figure 5 and summarized in Table 2). Lactosylceramide and gangliotriaosylceramide were recognized by all four strains, while only the 890 and 60611 strains bound to galactosylceramide. The most interesting difference was found for lactotetraosylceramide and neolactotetraosylceramide. Here, the 890 strain bound consistently to both compounds, the 60611 strain bound only to lactotetraosylceramide, the 68164 strain bound only to neolactotetraosylceramide, while the 19096 strain did not recognize any of these two glycosphingolipids.

### 3.6. Isolation and Characterization of an A. baumannii Binding Glycosphingolipid from Human Skin

In order to approach target cell *A. baumannii* binding glycosphingolipids, we next isolated glycosphingolipids from human skin. The major glycosphingolipids of normal human skin were characterized during the 1980s as glucosylceramide, lactosylceramide, globotriaosylceramide, globoside, and the gangliosides GM3 and GD3 [24]. However, a characterization with the methods available today has not been done. Here we isolated a total non-acid fraction from human skin and separated this into five subfractions, which were denoted fractions S-1–S-5 (Figure 6A, lanes 1–5).

Binding of *A. baumannii* to the five non-acid glycosphingolipid subfractions gave a distinct binding in the tetraosylceramide region of fraction S-4 (Figure 6B, lane 4). Chromatogram binding assays with the Galβ4GlcNAc-recognizing lectin from *E. crista-galli* [21] also gave a binding in the tetraosylceramide region of fraction S-4 (Figure 6C, lane 4), indicating the presence of neolactotetraosylceramide.

LC-ESI/MS of the oligosaccharides obtained from fraction S-4 by hydrolysis with endoglycoceramidase II gave a base chromatogram with molecular ion at *m/z* 706, demonstrating an oligosaccharide with one HexNAc and three Hex. A C-type fragment ion series (C_2_ at *m/z* 382 and C_3_ at *m/z* 544) identifying a Hex-HexNAc-Hex-Hex sequence was obtained by MS^2^ of the ion at *m/z* 706 (Figure 6D). Moreover, this spectrum had the characteristic ^0,2^A_2_-H_2_O fragment ion at *m/z* 263 and the ^0,2^A_3_ fragment ion at *m/z* 281, demonstrating a terminal Hex-HexNAc sequence with a 4-substitution of the HexNAc, i.e., a type 2 chain [17,22,23]. Thus, a neolacto tetrasaccharide (Galβ4GlcNAcβ3Galβ4Glc) was again identified.

### 3.7. Inhibition Studies

Preincubation of *A. baumannii* CCUG 890 with polyclonal antibodies against the N-terminal domain of CsuE (αEN) [5] did not affect the glycosphingolipid binding, i.e., binding to galactosylceramide, lactosylceramide, gangliotriaosylceramide, gangliotetraosylceramide, lactotetraosylceramide, and neolactotetraosylceramide was still obtained (exemplified in Figure 5F). Thus, the hydrophobic three-finger loops at the tip of the Csu pilus are not involved in *A. baumannii* glycosphingolipid binding.

## 4. Discussion

To establish infections, pathogenic bacteria need to adhere to host cells and tissues, and the majority of microbial attachment sites identified are glycoconjugates [13,14]. In the present study, the carbohydrate recognition by *A. baumannii* bacterial cells was characterized by binding of *A. baumannii* bacteria to glycosphingolipids on thin-layer chromatograms. Two *A. baumannii* binding glycosphingolipids of human and rabbit small intestine were isolated and characterized by mass spectrometry as neolactotetraosylceramide (Galβ4GlcNAcβ3Galβ4Glcβ1Cer) and lactotetraosylceramide (Galβ3GlcNAcβ3Galβ4Glcβ1Cer), respectively. Since *A. baumannii* also bound to reference lactotriaosylceramide (GlcNAcβ3Galβ4Glcβ1Cer), the basic binding element recognized is GlcNAc. Glycosphingolipids from human skin were isolated and *A. baumannii* binding neolactotetraosylceramide was characterized in this target tissue for the bacteria. In addition, as previously described for several other bacteria [14]. *A. baumannii* bound to galactosylceramide, lactosylceramide with phytosphingosine and/or hydroxy fatty acids, isoglobotriaosylceramide, gangliotriaosylceramide, and gangliotetraosylceramide. However, while the binding of *A. baumannii* to lactotriaosylceramide, neolactotetraosylceramide, and lactotetraosylceramide was very reproducible, binding to these compounds was only occasionally obtained. The detection limit for binding of *A. baumannii* to galactosylceramide gangliotriaosylceramide, neolactotetraosylceramide, and lactotetraosylceramide, estimated by binding to serial dilutions of glycosphingolipids on thin-layer chromatograms, was at the microgram level (1–4 µg). Thus, there was no clear preference for any of these glycosphingolipids.

Interestingly, the binding patterns obtained with the four *A. baumannii* strains were not entirely identical. All four strains bound to lactosylceramide and gangliotriaosylceramide, but only the 890 and 60611 strains bound to galactosylceramide. Furthermore, while the 890 strain bound consistently to both lactotetraosylceramide and neolactotetraosylceramide, the 60611 strain bound only to lactotetraosylceramide, the 68164 strain bound only to neolactotetraosylceramide, and the 19096 strain did not recognize either these glycosphingolipids. This indicates that the adhesins of the *A. baumannii* strains have differences in the architecture of their carbohydrate binding site(s).

The potential carbohydrate binding of *A. baumannii* CsuC–CsuE protein [5] was also tested during the initial binding experiments, but no glycosphingolipid binding was observed. Furthermore, the binding of *A. baumannii* to glycosphingolipids was not affected by preincubation with antibodies against the pilus tip protein CsuE. These antibodies, on the other hand, completely blocked *A. baumannii* biofilm formation [5]. Thus, the hydrophobic three-finger loops at the tip of the Csu pilus are essential for the formation of biofilms but are not involved in *A. baumannii* glycosphingolipid binding.

The ability of *A. baumannii* isolates to adhere to human epithelial cells has been investigated in cell culture experiments, and several proteins, e.g., the outer membrane protein A (OmpA) and the biofilm-associated protein (BAP), have been shown to play a role in the interaction of bacteria with human cells (reviewed in ref. [25]). Despite the multitude of candidate adhesins, no cellular ligands of these proteins have been characterized to date. Further studies of *A. baumannii* potential adhesins, including the adhesin(s) mediating glycosphingolipid binding, are thus needed.

Interestingly, there is a predicted *A. baumannii* protein termed lecA, which is described as a galactose binding protein the UniProt database (entry A0A1G5LY11). *A. baumannii* lecA has very high homology to the lecA (PA-IL) lectin of *Pseudomonas aeruginosa*, which preferentially binds to glycoconjugates with terminal Galα4Gal sequences and to some extent also to terminal Galα3Gal sequences [26]. Apart from isoglobotriaosylceramide, we did not observe binding of *A. baumannii* to any glycosphingolipids with terminal Galα4Gal and Galα3Gal sequences (e.g., globotriaosylceramide (no. 28 in Table 1) https://susy.mdpi.com/user/manuscripts/resubmit/0b5776af599a41dc5d4837e5ff40be13) or B penta (no. 12). However, in *P. aeruginosa* the lecA lectin is located mainly in the cytoplasm with only a small fraction on the cytoplasmic membrane [27]. If this is also the case with *A. baumannii* lecA, it might explain the absence of binding of the bacterial cells. Obviously the role of this putative lectin needs further investigations.

Binding of G fimbria of human uropathogenic *Escherichia coli*, and fimbriae belonging to the F17 family of bovine enterotoxigenic and invasive *E. coli* strains, to terminal GlcNAc has been reported, and the lectin domains of the G and F17 fimbriae have been co-crystallized with *N*-acetylglucosamine [28,29]. However, the target cell receptors for these fimbriae have not yet been identified.

The identification of the *A. baumannii* binding glycosphingolipids, lactotetraosylceramide, and neolactotetraosylceramide may further contribute to our understanding of the molecular mechanisms by which *A. baumannii* establish successful infection in human hosts and could hopefully guide the development of novel high-affinity ligands that may be used as anti-adhesive compounds against *A. baumannii* infections.

## Figures and Tables

**Figure 1 microorganisms-08-00612-f001:**
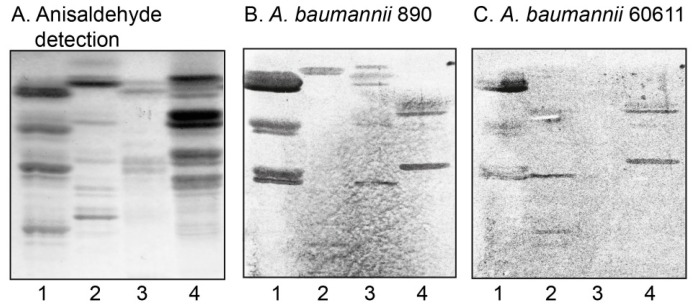
Screening for *Acinetobacter baumannii* carbohydrate recognition by binding to glycosphingolipids on thin-layer chromatograms. Chemical detection by anisaldehyde (**A**), and autoradiograms obtained by binding of *A. baumannii* strain CCUG 890 (**B**), and *A. baumannii* strain CCUG 60611 (**C**), followed by autoradiography for 12 h, as described under “Materials and Methods”. The solvent system used was chloroform/methanol/water (60:35:8, by volume). The lanes were: Lane 1, non-acid glycosphingolipids of rabbit small intestine, 40 μg; Lane 2, non-acid glycosphingolipids of rabbit thymus, 40 μg; Lane 3, non-acid glycosphingolipids of rat intestine, 40 μg; Lane 4, non-acid glycosphingolipids of human small intestine, 40 μg.

**Figure 2 microorganisms-08-00612-f002:**
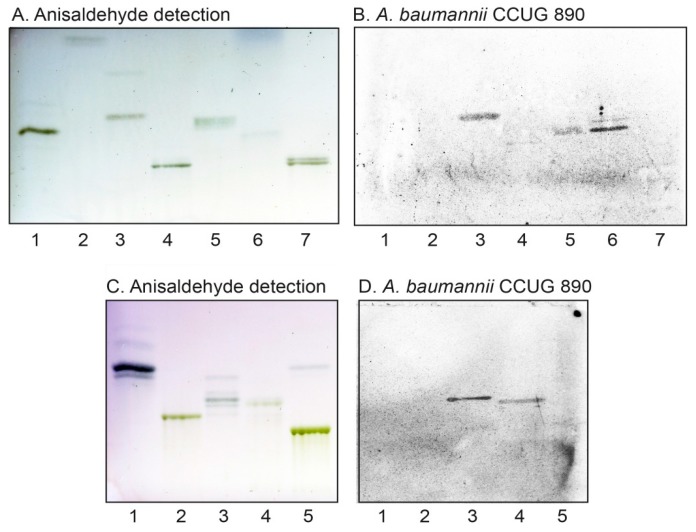
*A. baumannii* binding glycosphingolipids isolated from rabbit small intestine (fraction RI-4) and human small intestine (fraction HI-4). Chemical detection by anisaldehyde (**A**,**C**), and autoradiograms obtained by binding of *A. baumannii* strain CCUG 890 (**B**,**D**), followed by autoradiography for 12 h, as described under “Materials and Methods”. The solvent system used was chloroform/methanol/water (60:35:8, by volume). The lanes on A and B were: Lane 1, globotetraosylceramide (GalNAcβ3Galα4Galβ4Glcβ1Cer), 4 μg; Lane 2, galactosylceramide (Galβ1Cer), 4 μg; Lane 3, gangliotriaosylceramide (GalNAcβ4Galβ4Glcβ1Cer), 4 μg; Lane 4, Galili pentaosylceramide (Galα3Galβ4GlcNAcβ3Galβ4Glcβ1Cer), 4 μg; Lane 5, isoglobotriaosylceramide (Galα3Galβ4Glcβ1Cer), 4 μg; Lane 6, fraction RI-4 from rabbit small intestine, 4 μg; Lane 7, gangliotetraosylceramide (Galβ3GalNAcβ4Galβ4Glcβ1Cer), 4 μg. The lanes on (**C**,**D**) were: Lane 1, globotriaosylceramide (Galα4Galβ4Glcβ1Cer), 4 μg; Lane 2, H type 2 pentaosylceramide (Fucα2Galβ4GlcNAcβ3Galβ4Glcβ1Cer), 4 μg; Lane 3, lactotetraosylceramide (Galβ3GlcNAcβ3Galβ4Glcβ1Cer), 4 μg; Lane 4, fraction HI-4 from human small intestine, 4 μg; Lane 5, B type 2 hexaosylceramide (Galα3(Fucα2)Galβ4GlcNAcβ3Galβ4Glcβ1Cer), 4 μg.

**Figure 3 microorganisms-08-00612-f003:**
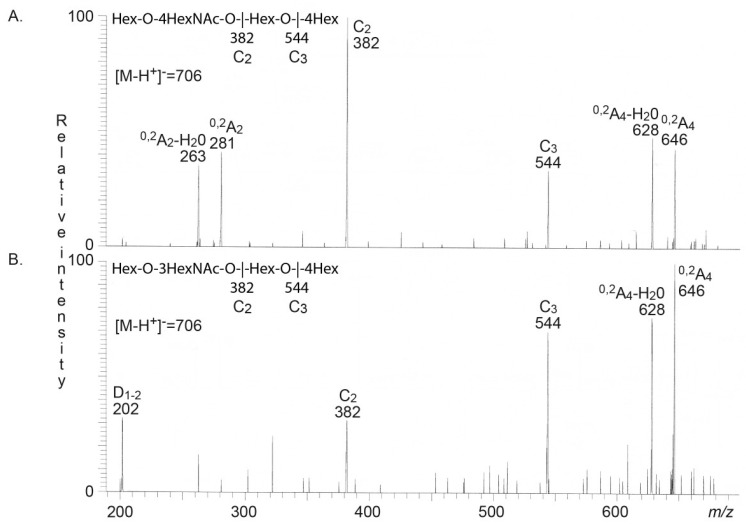
Characterization of the *A. baumannii* binding glycosphingolipids isolated from human and rabbit small intestine. (**A**) MS^2^ spectrum of the ion at *m/z* 706 (retention time 21.4 min) from liquid chromatography–electrospray ionization–mass spectrometry (LC-ESI/MS) of the oligosaccharides derived from the *A. baumannii* binding glycosphingolipid fraction HI-4 from human small intestine by digestion with *Rhodococcus* endoglycoceramidase II. The interpretation formula shows the deduced oligosaccharide sequence. (**B**) MS^2^ spectrum of the ion at *m/z* 706 (retention time 21.1 min) from LC-ESI/MS of the oligosaccharides derived from the *A. baumannii* binding glycosphingolipid fraction RI-4 from rabbit small intestine by digestion with *Rhodococcus* endoglycoceramidase II. The interpretation formula shows the deduced oligosaccharide sequence.

**Figure 4 microorganisms-08-00612-f004:**
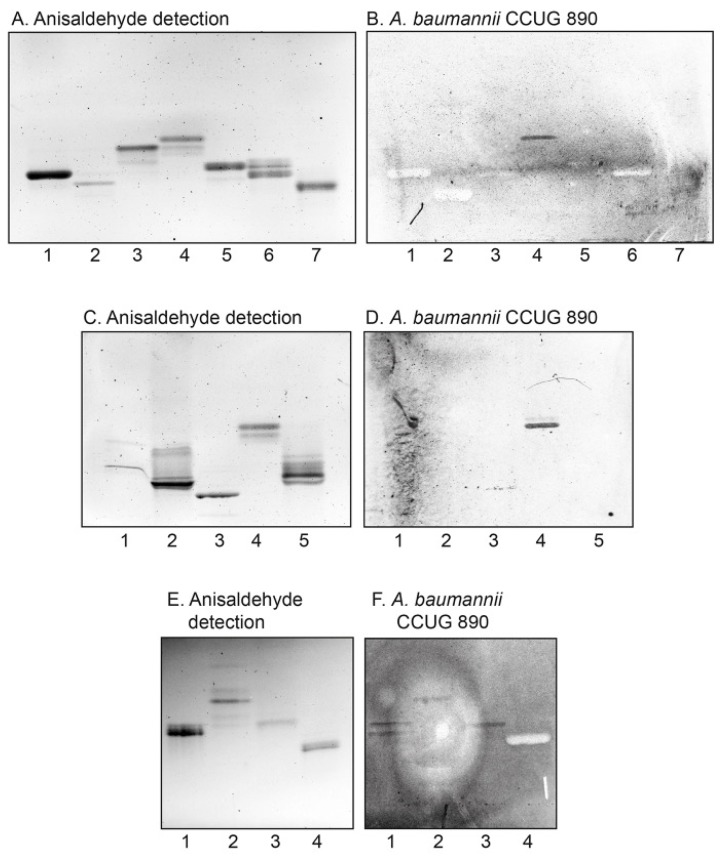
Binding of *A. baumannii* to reference glycosphingolipids on thin-layer chromatograms. Thin-layer chromatograms stained with anisaldehyde (**A**,**C**,**E**) and autoradiograms obtained by binding of ^35^S-labeled *A. baumannii* strain CCUG 890 (**B**,**D**,**F**), followed by autoradiography for 12 h, as described under “Materials and Methods”. The solvent system used was chloroform/methanol/water (60:35:8, by volume). The lanes on (**A**,**B**) were: Lane 1, H type 2 pentaosylceramide (Fucα2Galβ4GlcNAcβ3Galβ4Glcβ1Cer), 4 μg; Lane 2, B type 2 hexaosylceramide (Galα3(Fucα2)Galβ4GlcNAcβ3Galβ4Glcβ1Cer), 2 μg; Lane 3, globotetraosylceramide (GalNAcβ3Galα4Galβ4Glcβ1Cer), 4 μg; Lane 4, lactotetraosylceramide (Galβ3GlcNAcβ3Galβ4Glcβ1Cer), 4 μg; Lane 5, H type 1 pentaosylceramide (Fucα2Galβ4GlcNAcβ3Galβ4Glcβ1Cer), 4 μg; Lane 6, Le^a^ pentaosylceramide (Galβ3(Fucα4)GlcNAcβ3Galβ4Glcβ1Cer), 4 μg; Lane 7, Le^b^ hexaosylceramide (Fucα2Galβ3(Fucα4)GlcNAcβ3Galβ4Glcβ1Cer), 4 μg. The lanes on (**C**,**D**) were: Lane 1, NeuAcα3neolactotetraosylceramide (NeuAcα3-Galβ4GlcNAcβ3Galβ4Glcβ1Cer), 2 μg; Lane 2, NeuGcα3neolactotetraosylceramide (NeuGcα3Galβ4GlcNAcβ3Galβ4Glcβ1Cer), 4 μg; Lane 3, NeuGcα3neolactohexa-osylceramide (NeuGcα3Galβ4GlcNAcβ3Galβ4GlcNAcβ3Galβ4Glcβ1Cer), 4 μg; Lane 4, lactotetraosylceramide (Galβ3GlcNAcβ3Galβ4Glcβ1Cer), 4 μg; Lane 5, NeuAcα3lactotetraosylceramide (NeuAcα3Galβ3GlcNAcβ3Galβ4Glcβ1Cer), 4 μg. The lanes on (**E**,**F**) were: Lane 1, neolactotetraosylceramide (Galβ4GlcNAcβ3Galβ4Glcβ1Cer), 4 μg; Lane 2, lactotriaosylceramide (GlcNAcβ3Galβ4Glcβ1Cer), 4 μg; Lane 3, lactotetraosylceramide (Galβ3GlcNAcβ3Galβ4Glcβ1Cer), 4 μg; Lane 4, H type 2 pentaosylceramide (Fucα2Galβ4GlcNAcβ3Galβ4Glcβ1Cer), 4 μg.

**Figure 5 microorganisms-08-00612-f005:**
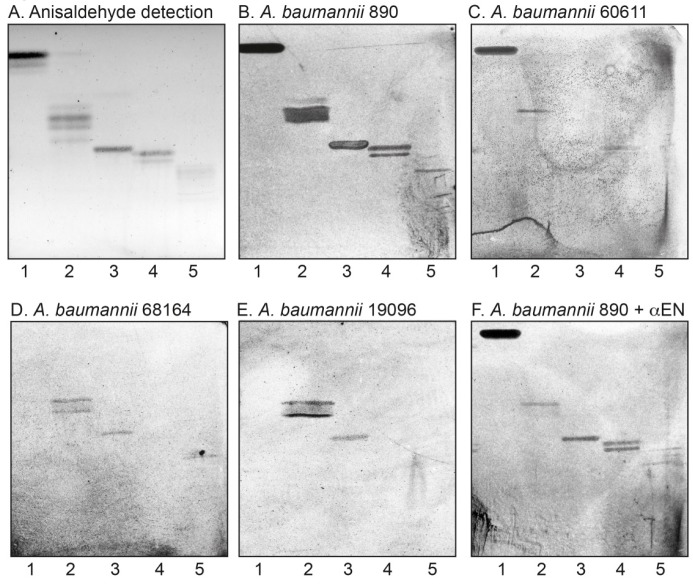
Binding of *A. baumannii* to glycosphingolipids on thin-layer chromatograms. Chemical detection by anisaldehyde (**A**) and autoradiograms obtained by binding of *A. baumannii* strain CCUG 890 (**B**), *A. baumannii* strain CCUG 60611 (**C**), *A. baumannii* strain CCUG 68164 (**D**), *A. baumannii* strain CCUG 19096 (**E**), and *A. baumannii* strain CCUG 890 incubated with antibodies raised against the N-terminal domain of CsuE (αEN) (**F**), followed by autoradiography for 12 h, as described under “Materials and Methods”. The solvent system used was chloroform/methanol/water (60:35:8, by volume). The lanes were: Lane 1, galactosylceramide (Galβ1Cer), 4 μg; Lane 2, lactosylceramide (Galβ4Glcβ1Cer) with hydroxy ceramide, 4 μg; Lane 3, gangliotriaosylceramide (GalNAcβ4Galβ4Glcβ1Cer); Lane 4, lactotetraosylceramide (Galβ3GlcNAcβ3-Galβ4Glcβ1Cer), 2 μg; Lane 5, neolactotetraosylceramide (Galβ4GlcNAcβ3Galβ4Glcβ1Cer), 2 μg.

**Figure 6 microorganisms-08-00612-f006:**
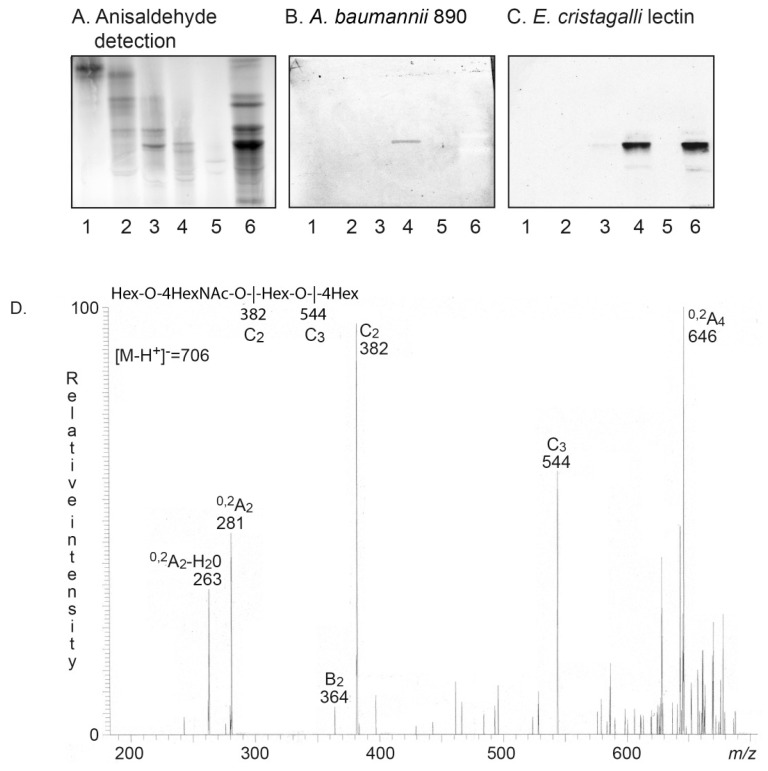
Non-acid glycosphingolipids of human skin: Characterization of neolactotetraosylceramide. (**A**) Chemical detection by anisaldehyde. (**B**) Autoradiogram obtained by binding of *A. baumannii* strain CCUG 890. (**C**) Autoradiogram obtained by binding of Galβ4GlcNAc-recognizing lectin from *Erythrina crista-galli.* The lanes were: Lanes 1–5, fractions S-1–S-5 isolated from human skin, 4 μg/lane; Lane 6, reference non-acid glycosphingolipids of human erythrocytes blood group AB, 40 μg. (**D**) MS^2^ spectrum of the ion at *m/z* 706 (retention time 21.0 min) from LC-ESI/MS of the oligosaccharides derived from fraction S-4 from human skin by digestion with *Rhodococcus* endoglycoceramidase II. The interpretation formula shows the deduced oligosaccharide sequence.

**Table 1 microorganisms-08-00612-t001:** Binding of *Acinetobacter baumannii* to reference glycosphingolipids on thin-layer chromatograms.

No.	Trivial Name	Structure	Binding ^a^	Source
Simple compounds				
1.	GalCer (d18:1-h24:0) ^b^	Galβ1Cer	(+)	Porcine intestine
2.	Sulfatide	SO_3_-Galβ1Cer	-	Porcine intestine
3.	LacCer (d18:1-16:0-24:0)	Galβ4Glcβ1Cer	-	Human neutrophils
4.	LacCer (t18:0-h16:0-h24:0)	Galβ4Glcβ1Cer	+	Dog intestine
Ganglioseries				
5.	Gangliotri	GalNAcβ4Galβ4Glcβ1Cer	(+)	Guinea pig erythrocytes
6.	Gangliotetra	Galβ3GalNAcβ4Galβ4Glcβ1Cer	(+)	Mouse intestinee
7.	GD1a	NeuAcα3Galβ3GalNAcβ4(NeuAcα3)Galβ4Glcβ1Cer	-	Human brain
Neolactoseries				
8.	Lactotri	GlcNAcβ3Galβ4Glcβ1Cer	+	Porcine intestine
9.	Neolactotetra	Galβ4GlcNAcβ3Galβ4Glcβ1Cer	+	Human neutrophils
10.	H type 2 penta	Fucα2Galβ4GlcNAcβ3Galβ4Glcβ1Cer	-	Human erythrocytes
11.	Le^x^ penta	Galβ4(Fucα3)GlcNAcβ3Galβ4Glcβ1Cer	-	Dog intestine
12.	B penta	Galα3Galβ4GlcNAcβ3Galβ4Glcβ1Cer	-	Rabbit erythrocytes
13.	Le^y^ hexa	Fucα2Galβ4(Fucα3)GlcNAcβ3Galβ4Glcβ1Cer	-	Dog intestine
14.	B type 2 hexa	Galα3(Fucα2)Galβ4GlcNAcβ3Galβ4Glcβ1Cer	-	Human erythrocytes
15.	B type 2 hepta	Galα3(Fucα2)Galβ4(Fucα3)GlcNAcβ3Galβ4Glcβ1Cer	-	Human erythrocytes
16.	A type 2 hepta	GalNAcα3(Fucα2)Galβ4(Fucα3)GlcNAcβ3Galβ4Glcβ1Cer	-	Human erythrocytes
17.	NeuAcα3-neolactotetra	NeuAcα3Galβ4GlcNAcβ3Galβ4Glcβ1Cer	-	Human erythrocytes
18.	NeuAcα6-neolactotetra	NeuAcα6Galβ4GlcNAcβ3Galβ4Glcβ1Cer	-	Human meconium
19.	NeuGcα3-neolactotetra	NeuGcα3Galβ4GlcNAcβ3Galβ4Glcβ1Cer	-	Goat erythrocytes
20.	NeuGcα3-neolactohexa	NeuGcα3Galβ4GlcNAcβ3Galβ4GlcNAcβ3Galβ4Glcβ1Cer	-	Rabbit thymus
Lactoseries				
21.	Lactotetra	Galβ3GlcNAcβ3Galβ4Glcβ1Cer	+	Human meconium
22.	Le^a^ penta	Galβ3(Fucα4)GlcNAcβ3Galβ4Glcβ1Cer	-	Human intestine
23.	Le^b^ hexa	Fucα2Galβ3(Fucα4)GlcNAcβ3Galβ4Glcβ1Cer	-	Human meconium
24.	NeuAc-lactotetra	NeuAcα3Galβ3GlcNAcβ3Galβ4Glcβ1Cer	-	Human pancreas cancer
Globoseries				
25.	Isoglobotri	Galα3Galβ4Glcβ1Cer	(+)	Dog intestine
26.	Isoglobotetra	GalNAcβ3Galα3Galβ4Glcβ1Cer	(+)	Rat intestine
27.	Galα3-isoglobotri	Galα3Galα3Galβ4Glcβ1Cer	(+)	Cat intestine
28.	Globotri	Galα4Galβ4Glcβ1Cer	-	Human erythrocytes
29.	Globotetra	GalNAcβ3Galα4Galβ4Glcβ1Cer	-	Human erythrocytes

^a^ Binding is defined as follows: + denotes a highly reproducible binding of *A. baumannii* CCUG 890 when 4 µg of the glycosphingolipid was applied on the thin-layer chromatogram, while (+) denotes an occasional binding, and - denotes no binding at 4 µg. ^b^ In the shorthand nomenclature for fatty acids and bases, the number before the colon refers to the carbon chain length and the number after the colon gives the total number of double bonds in the molecule. Fatty acids with a 2-hydroxy group are denoted by the prefix h before the abbreviation, e.g., h16:0. For long chain bases, d denotes dihydroxy and t trihydroxy. Thus, d18:1 designates sphingosine (1,3-dihydroxy-2-aminooctadecene) and t18:0. phytosphingosine (1,3,4-trihydroxy-2-aminooctadecene).

**Table 2 microorganisms-08-00612-t002:** Comparison of glycosphingolipid binding of *Acinetobacter baumannii* strains.

No.	Trivial name	Structure	CCUG 19096	CCUG 890	CCUG 60611	CCUG 68614
1.	GalCer (d18:1-h24:0) ^b^	Galβ1Cer	− ^a^	+	+	−
2.	LacCer (t18:0-h16:0-h24:0)	Galβ4Glcβ1Cer	+	+	+	+
3.	Gangliotri	GalNAcβ4Galβ4Glcβ1Cer	+	+	+	+
4.	Neolactotetra	Galβ4GlcNAcβ3Galβ4Glcβ1Cer	−	+	−	+
5.	Lactotetra	Galβ3GlcNAcβ3Galβ4Glcβ1Cer	−	+	+	−

^a^ Binding is defined as follows: + denotes a highly reproducible binding of *A. baumannii* when 4 µg of the glycosphingolipid was applied on the thin-layer chromatogram, while − denotes no binding at 4 µg. ^b^ In the shorthand nomenclature for fatty acids and bases, the number before the colon refers to the carbon chain length and the number after the colon gives the total number of double bonds in the molecule. Fatty acids with a 2-hydroxy group are denoted by the prefix h before the abbreviation, e.g., h16:0. For long chain bases, d denotes dihydroxy and t trihydroxy. Thus d18:1 designates sphingosine (1,3-dihydroxy-2-aminooctadecene) and t18:0 phytosphingosine (1,3,4-trihydroxy-2-aminooctadecene).

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
