# Peer review of "Glycosphingolipids Recognized by Acinetobacter baumannii"

_microorganisms, 2020, doi:10.3390/microorganisms8040612_

Round 1

Reviewer 1 Report

Miralda et al. described the different glycosphingolipids that can be recognized by A. baumannii on thin-layer chromatograms. The binding-active glycosphingolipids were identified as neolactotetraosylceramide and lactotetraosylceramide with mass spectrometry and further binding assay. The investigators provided clear evidence to demonstrate the interactions between several types of glycosphingolipids and different strains of A. baumannii. It is very interesting why these strains bind to the specific glycosphingolipids (Page 8, line 305-308), which may deserve further investigation. Also, I list several comments for the authors' consideration as below.

Did they submit their raw data of mass spectrometry to the public datasets?

Page 11, line 353. Why did not the authors show the related data? The inhibition results are critical for evaluating the glycosphingolipid binding in this study. Their conclusions will be more convincing if they can include these inhibition studies.

Table 1 can be improved because some features are shown incorrectly or incompletely.

Author Response

We thank the reviewer for the constructive criticism of our manuscript which has improved the manuscript. The points made by the reviewer have now been addressed, and we have made the following changes in the manuscript. 

  1. It is very interesting why these strains bind to the specific glycosphingolipids (Page 8, line 305-308), which may deserve further investigation.

We also find this highly interesting and aim to investigate this further. First we will try to identify the glycosphingolipid binding adhesins – we will do knock-out of a number of outer membrane proteins in the “multi-binding” strain 890 and investigate the effects on glycolipid binding. We will also attempt affinity isolation of membrane proteins. However, this will the subject of a separate report.

  1. Did they submit their raw data of mass spectrometry to the public datasets?

My colleagues at the Department of Biomedicine are co-authors of the recent article about submission of MS-based glycomics information into the public repository UniCarb-DR (Rojas-Macias et al. Nat Commun. 2019 10, 3275. doi: 10.1038/s41467-019-11131-x), and they are currently transferring all our MSdata files from glycosphingolipid-derived oligosaccharides to the UniCarb-DR database. So this is underway.

  1. Page 11, line 353. Why did not the authors show the related data? The inhibition results are critical for evaluating the glycosphingolipid binding in this study. Their conclusions will be more convincing if they can include these inhibition studies.

Binding of A. baumannii strain 890 preincubated antibodies raised against the N-terminal domain of CsuE is now shown in Fig. 5F.

  1. Table 1 can be improved because some features are shown incorrectly or incompletely.

We apologize for this mess. Table 1 has been revised.

Reviewer 2 Report

The manuscript is well-written and the results are reported in a clear way. However, some issues were found during the revision process, and in my opinion they should be carefully re-checked by the authors prior to a further consideration.

Introduction

  • Lines 65-72: In this part you are introducing the results. I suggest to delete it and leave only the aim of the work.

Methods

  • Line 87: has been NMR used for characterization? No NMR results are reported below. If the technique was not employed, please delete it from the methods. Otherwise, report some results.
  • Line 145: which is the gradient used? Please specify.

Results

  • In this work, the binding affinity of A. baumannii to the different glycosphingolipids is not reported. However, it would be an important data to include among the results. Was this calculated? If not, is it possible to calculate it and add the data to the results?
  • Line 192: Why did you not report a figure of the designated fraction HI-4, but only of the RI-4?
  • Lines 248-250: “characterized” should be changed with “tentatively identified”, as you are not using a comparison with reference standards.

Discussion

  • Lines 367-369: “A. baumannii also occasionally bound to galactosylceramide, lactosylceramide with phytosphingosine and/or hydroxy fatty acids, isoglobotriaosylceramide, gangliotriaosylceramide and gangliotetraosylceramide”. What does mean “occasionally”? Was a binding affinity calculated? As already commented for the results part, this would be an important data to support these observations, as well as the conclusions of the work.

Minor issues:

  • Table 1 is not properly reported, and the name of the compounds appear incomplete.
  • Explain the meaning of each abbreviation at its first appearance in the text.
  • Check the appropriateness of the spelling and grammar throughout the text.

Author Response

We thank the reviewer for the constructive criticism of our manuscript which has improved the manuscript. The points made by the reviewer have now been addressed, and we have made the following changes in the manuscript. 

  1. Lines 65-72: In this part you are introducing the results. I suggest to delete it and leave only the aim of the work.

This part has now been deleted.

  1. Line 87: has been NMR used for characterization? No NMR results are reported below. If the technique was not employed, please delete it from the methods. Otherwise, report some results.

Several of our reference glycosphingolipids were isolated and characterized 10-20 years, and at that time mass spectrometry plus proton NMR was used for structural characterization. Therefore NMR is mentioned here. However, structural characterization in the present study was done by mass spectrometry and lectin binding.

  1. Line 145: which is the gradient used? Please specify.

This information has been added (page 7, lines 11-13).

  1. In this work, the binding affinity of A. baumannii to the different glycosphingolipids is not reported. However, it would be an important data to include among the results. Was this calculated? If not, is it possible to calculate it and add the data to the results?

The relative affinity of A. baumannii for the binding-active glycosphingolipids was estimated by binding to serial dilutions of glycosphingolipids on thin-layer chromatograms. In this assay the bacteria bound to gangliotriaosylceramide, neolactotetraosylceramide and lactotetraosylceramide with detection limit of 1-2 mg, while galactosylceramide required 4 mg. Thus, there was no clear preference for any of these glycosphingolipids. This information is now given on page 16, lines 17-22.

  1. Line 192: Why did you not report a figure of the designated fraction HI-4, but only of the RI-4?

Fraction HI-4 is now shown in Fig. 2C and D, lane 4.

  1. Lines 248-250: “characterized” should be changed with “tentatively identified”, as you are not using a comparison with reference standards.

This has been corrected.

  1. Lines 367-369: “A. baumannii also occasionally bound to galactosylceramide, lactosylceramide with phytosphingosine and/or hydroxy fatty acids, isoglobotriaosylceramide, gangliotriaosylceramide and gangliotetraosylceramide”. What does mean “occasionally”?

While the binding of A. baumannii to lactotriaosylceramide, neolactotetraosylceramide and lactotetraosylceramide was highly reproducible, binding to galactosylceramide, lactosylceramide with phytosphingosine and/or hydroxy fatty acids, isoglobotriaosylceramide, gangliotriaosylceramide and gangliotetraosylceramide was sometimes absent.

We have re-formulated this part (page 20, lines 18-20), and also the corresponding section under Results page 16, lines 4-10, in order to clarify this.

  1. Was a binding affinity calculated? As already commented for the results part, this would be an important data to support these observations, as well as the conclusions of the work.

We could not find any preferential binding to any of the binding-active glycolipids. When binding to serial dilutions of glycosphingolipids on thin-layer chromatograms the detection limit for binding of A. baumannii to galactosylceramide gangliotriaosylceramide, neolactotetraosylceramide and lactotetraosylceramide was at the microgram level (1-4 microgram). This is now also mentioned under Discussion (page 20, lines 20-24).

  1. Table 1 is not properly reported, and the name of the compounds appear incomplete.

We apologize for this mess. Table 1 has been revised.

  1. Explain the meaning of each abbreviation at its first appearance in the text.

This has been corrected.

  1. Check the appropriateness of the spelling and grammar throughout the text.

We have corrected all the mistakes we could find and apologize.

Round 2

Reviewer 2 Report

The authors appropriately addressed the issues that were encountered during the revision process, and the manuscript was modified accordingly. I suggest this manuscript for publication in its actual form.

Author Response

We have now made an addition to the Discussion (page 21, lines 12-24), and sincerely hope that this will allow a final acceptance of our manuscript.
